# Development of a Tellurium Speciation Study Using IC-ICP-MS on Soil Samples Taken from an Area Associated with the Storage, Processing, and Recovery of Electrowaste

**DOI:** 10.3390/molecules26092651

**Published:** 2021-04-30

**Authors:** Katarzyna Grygoyć, Magdalena Jabłońska-Czapla

**Affiliations:** Institute of Environmental Engineering Polish Academy of Sciences, M. Sklodowskiej-Curie 34 St., 41-819 Zabrze, Poland; katarzyna.grygoyc@ipispan.edu.pl

**Keywords:** tellurium, technology critical element, TCE, BCR, speciation, IC-ICP-MS, ICP-MS, fractionation, WEEE, electrowaste

## Abstract

The optimization and validation of a methodology for determining and extracting inorganic ionic Te(VI) and Te(IV) forms in easily-leached fractions of soil by Ion Chromatography-Inductively Coupled Plasma-Mass Spectrometry (IC-ICP-MS) were studied. In this paper, the total concentration of Te, pH, and red-ox potential were determined. Ions were successfully separated in 4 min on a Hamilton PRPX100 column with 0.002 mg/kg and 0.004 mg/kg limits of detection for Te(VI) and Te(IV), respectively. Soil samples were collected from areas subjected to the influence of an electrowaste processing and sorting plant. Sequential chemical extraction of soils showed that tellurium was bound mainly with sulphides, organic matter, and silicates. Optimization of soil extraction allowed 20% average extraction efficiency to be obtained, using 100 mM citric acid as the extractant. In the tested soil samples, both tellurium species were present. In most cases, the soils contained a reduced Te form, or the concentrations of both species were similar.

## 1. Introduction

Global trade networks of goods, in which materials move along the value chain of mining, processing, manufacture, use, disposal, collection, and waste management, have increased in complexity in recent years as multiple countries are involved in the life-cycles of products [1]. As a consequence of the growing use of electronic and industrial products, increasing amounts of TCE (Technology Critical Elements) are being released to the environment. The growing supply of TCE to the environment requires an understanding of their mobility, reactivity, and chemical transformations in the environment, which are critically dependent on their chemical form (speciation). The total concentrations of TCE elements are at trace levels, which make their speciation analysis challenging. One of the Less Studied Technology Critical Element (LSTCE) is tellurium, which is increasingly employed in optical magnetic discs, solar panels, and Peltier device production [2].

Tellurium is one of the chalcophile elements that belong to group 16 in the Periodic Table. Its chemical behavior is similar to selenium and in the natural environment can exist in several redox states: telluride (−II), elemental tellurium (0), tellurite (IV) and tellurate (VI) both in organic and inorganic forms. Although chemical properties of tellurium are well known there is few information about its chemical behavior in natural systems. Under ordinary environmental conditions (oxic systems, circumneutral pH, absence of ligands other than those derived from water), the dominant species should be those of Te(IV), while Te(VI) species will be potent oxidants [3]. Te(IV)/Te(VI) ratio can differ under different environmental and biological conditions but studies in natural systems are lacking due to extremely low Te concentrations in geological, environmental, and biological samples and most of them is focused on developing the research methodology and rather concerns the total content of tellurium. Research of Hai-Bo Qin et al. [4] showed that in abandoned mine tailings contaminated soil Te was present as a mixture of Te(VI) and Te(IV) species, and Fe(III) hydroxides were the host phases for Te(IV), and Te(VI), but Te(IV) could be also retained by illite. The values for total tellurium in soil and sediments vary in general from less than one ppb to a few ppb (µg/kg) depending on locations and sources of contamination [5]. Te content in some types in soils from USA was in the range 0.02 and 0.69 mg/kg [6]. Ferri et al. [7] studied tellurium species concentration in Soil NIST SRM 2709 (National Institute of Standards and Technology, Standard Reference Material) and showed that in this material the content of inorganic tellurium species was 50% Te(VI) and Te(IV). Harada and Takahashi [8] studied the distribution and speciation of Te between the solid and aqueous phases in synthetic soil. Under oxic conditions Te was mainly associated with iron (III) hydroxides, and Te(IV) and Te(VI) species were both found to inner-sphere complexes.

Tellurium is considered toxic and teratogenic, and there are indications that tellurite, Te(IV), can be more toxic than tellurate, Te(VI) [9]. Because of easy phyto- and bioaccumulation, local tellurium pollution can pose a threat [10]. Waste electronic equipment, in the form of electrowaste (e-waste), is often the source of precious metals belonging to TCE [2,10,11]. E-waste is increasingly flooding the world, and is one of the fastest growing waste streams in the world in terms of volume and its environmental impact on the planet. Moreover, it has garnered significant popular and academic attention since 2010 [12].

The term speciation is often used to define an analytical procedure conducted to determine and identify chemical species and to measure their distribution in a particular sample or matrix. Chemical speciation can also be seen as the process of identifying and determining specific chemical species to learn about the availability and mobility of metals in the context of understanding their chemical behavior. Speciation defines chemical forms, whereas fractionation classifies analytes or analyte groups in a sample with regard to their physical and chemical properties [13]. The results of toxicological tests demonstrate that specific element forms, rather than the total element content, often have a profound impact on living organisms. For that reason, information about the content of different forms of the element occurrence is more important than knowing the total element content in the sample. Generally, it is believed that elements in ionic forms demonstrate biological activity and toxicity to living organisms. It is necessary to integrate separation and spectroscopic techniques, or combinations of techniques, to measure the ionic element forms [14]. One of the ways tellurium speciation in environmental samples is the use of the solid-phase extraction methods. Compared to hyphenated separation techniques, selective solid-phase extraction methods are interesting alternatives for samples with lower concentrations [15,16,17].

Hyphenated techniques combining a high resolution chromatographic separation with ultrasensitive element-specific detection by inductively coupled plasma mass spectrometry (ICP-MS) often offer the unique possibility of accessing the information on the identities and concentrations of individual metal species present in environmental samples. Unfortunately, the data on the speciation of TCE in the literature are relatively scarce and essentially limited to Ge, Te, Tl, Pt, Pd, Rh, and Gd [18]. In tellurium speciation, as in the case of other ultra-trace elements, combined techniques are dominant. One such technique is headspace single-drop microextraction (HS-SDME) combined with graphite furnace atomic absorption spectrometry (GFAAS) [19]. Another technique used in tellurium speciation is dispersive liquid–liquid microextraction combined with electrothermal atomic absorption spectrometry (DLLME) [20]. This speciation was based on the selective complex formation of APDC (ammonium pyrrolidine dithiocarbamate) as a chelating agent with Te(VI). The concentration of Te(IV) was calculated as the difference between the total tellurium and Te(VI) concentrations [20]. The best solution is to use techniques that allow simultaneous determination of Te(IV) and Te(VI), which allows for calculate a Te mass balance if the total content is measured separately. One such technique is ion chromatography-hydride generation-atomic fluorescence spectrometry (LC-HG-AFS) [21]. Anion-exchange chromatography with complexing agents in the mobile phase, ethylenediaminetetraacetic acid (EDTA) and potassium hydrogen phthalate (KHP), was used for separating Te(VI) and Te(IV). In the literature, we can find a few examples of tellurium speciation where atomic absorption spectrometry (AAS) or atomic fluorescence spectroscopy (AFS) are used to detect one form of tellurium and the total content of this element [22,23,24,25,26].

The application of hyphenated techniques such as HPLC-ICP-MS or IC-ICP-MS allows for speciation analysis and simultaneous determination of several ionic forms of elements [27,28]. So far, this combined technique has not been applied to research on tellurium speciation in soils. Direct determination of tellurium in geological samples by inductively coupled plasma mass spectrometry (ICP-MS) is often complicated by its low abundance, poor analytical sensitivity, and the presence of xenon interferences. Fortunately, accurate results, with low xenon interferences, can be obtained in the majority of matrices using both the ^125^Te and ^126^Te isotopes. Thanks to this, it is possible to combine analytical techniques in tellurium speciation studies using ICP-MS, with excellent sensitivity and ultratrace amounts of tellurium detection [29,30,31,32,33], for example 0.56 ng/L when analyzing Te(IV) in water [27]. There are very few publications on the speciation of tellurium in soils [8], and in the available literature, many authors have only studied the total content of tellurium in soils, or only one of the species [34].

The study verified and validated the methodology of tellurium extraction as well as simultaneous determination of two tellurium species, Te(VI) and Te(IV), in soils from the areas surrounding the Waste Electric and Electronic Equipment (WEEE) processing and sorting plant.

## 2. Results and Discussion

### 2.1. Optimization of Tellurium Speciation

#### 2.1.1. Elution Optimization

During the research on separating tellurium ionic forms, the type of column (Hamilton PRP-X100 and Dionex IonPac AS7), concentration of eluents (8–10 mM Na_2_EDTA and 2–6 mM KHP), pH of eluents (in the range of 4.29–4.32), separation temperature (20–30 °C), method of preparation of the standards (water or 0.5% HNO_3_), and complexing with complexing acids were optimized. Te(IV)/Te(VI) ions were separated with the Hamilton PRP-X100 column (150 mm × 4.6 mm, 5 µm). The anion exchange column Dionex IonPac AS7 (250 mm × 4 mm, 10 µm) was tested, but the obtained chromatogram presented wider peaks and stronger tailings of tellurium species (data not shown).

#### 2.1.2. The Influence of the Complexing Reagents on Tellurium Species Separation

Even the use of two complexing reagents present in the eluent, such as Na_2_EDTA and KHP, did not separate the two forms. It was necessary to use an additional complexing agent so as to keep one of the forms longer on the column. The effect of three acids (citric acid, tartaric acid, oxalic acid) as a factor complexing tellurium ions was tested, analogous to the work of Zheng et al. [35], in which the effect of complexing factors on antimony speciation was studied. Te(IV) and Te(VI) at a concentration of 1 g/L were diluted with 50 mM solution of citric, tartaric, and oxalic acids. Then, 15 min were allowed for complexing and the standards were injected into a Hamilton PRP-X100 column using 10 mM Na_2_EDTA and 6 mM KHP as eluent. In the case of tartaric acid, the Te(IV) peak was wide and tailing, and its intensity was lower compared to the peak when citric acid was used. When oxalic acid was used as the complexing agent, one broad peak was obtained; this reagent was not suitable for Te(IV) complexing. Therefore, each standard and real sample was prepared by the addition of 50 mM citric acid.

#### 2.1.3. Optimization of the Chromatographic Separation

Initially, attempts were made to separate Te(IV) and Te(VI) using 8 mM EDTA (ethylenediaminetetraacetic acid) and 2 mM KHP (potassium hydrogen phthalate) as eluent based on work by Vinas et al. [21]. However, no proper separation was obtained. Only the application of Na_2_EDTA (disodium ethylenediaminetetraacetic acid) together with KHP allowed the separation of both ionic tellurium forms. Variable concentrations of KHP (2, 4, 6 mM) and Na_2_EDTA (8 and 10 mM) in eluent were tested. The use of higher KHP concentration shortened the retention time of the Te(IV) peak. However, the increase in the KHP concentration did not affect the retention time of the Te(VI) peak. The use of phthalic acid instead of KHP was also tested along with 10 mM Na_2_EDTA (pH = 4.0), but the Te(IV) peak had a greater retention time, was wider, and exhibited tailing. The most optimal separation conditions were obtained using 10 mM Na_2_EDTA together with 6 mM KHP at pH in the range of 4.29–4.32 as eluent.

#### 2.1.4. Preparation of Standards

Single-element tellurium standards were prepared daily. The 1g/L Te(VI) and 100 mg/L Te(IV) standards were stored in the fridge at 4 °C. The Te(IV) standard, due to the fact that it contains some Te(VI), was reduced by the addition of 5 mL concentrated HCl at 80 °C for 30 min in a water bath. The 1 g/L of tellurium(VI) standard was prepared in advance, stored in the fridge at 4 °C, and was stable in a 0.5% HNO_3_ solution as tested. Standards were diluted with 50 mM citric acid. The standards of tellurium species were mixed together during calibration preparation. Calibration curves were obtained with measurement of 1 μg/L, 10 μg/L, and 25 μg/L standard solutions for Te(IV) and Te(VI), respectively. A linear model of the dependence of concentration of the total number of analyte counts was selected. The coefficient of determination of calibration curves R^2^ was between 0.9996 and 0.9901. Figure 1 presents superimposed chromatograms obtained after analyzing Te standard solutions, and Figure 2 shows calibration curves of tellurium species. Selected separation parameters are presented in Table 1. On the other hand, Figure 3 shows the chromatogram of the real soil sample 70 Edel and the chromatogram of standard both tellurium species with a 10 μg/L concentration.

#### 2.1.5. Optimization of Soil Extraction for the IC-ICP-MS Analysis

The extraction efficiency of soils for the tellurium species content by shaking was insufficient, and it was decided to change the extraction method. It was chosen to use ultrasound to improve the degree of tellurium extraction from the soil. The effect of extraction time on the degree of tellurium leaching after 1, 2, 3, and 4 h was investigated.

Figure 4 shows results of tellurium extraction from soils using various solutions (100 mM citric acid, 20 mM Na_2_EDTA, 300 mM ammonium tartrate, and a mixture of 100 mM citric acid with 20 mM Na_2_EDTA pH = 3.8). The results showed that the best extraction efficiency was obtained by using 100 mM citric acid as an extractant that washes out tellurium from soils within 4 h. The extraction efficiency of the soil samples varied significantly from 9% for the 110 Edel soil sample, up to 47% for the 52 Edel sample (Table 2). The extracts of these soils differed significantly, even organoleptically. Forest soil extracts were dark brown in color, and those from urban areas were straw-colored. e.g., soil samples with low extraction efficiency (105 Edel,107 Edel, 110 Edel, 112 Edel) were typically forest soils, containing a lot of organic substances, and their extracts were dark brown in color. On the other hand, the sample extracts, e.g., 52 Edel, 56 Edel, 58 Edel, 61 Edel had a straw color, and the extraction efficiency of these soils was high.

#### 2.1.6. Sequential Chemical Extraction

Sequential chemical extraction of the studied soils showed (Figure 5) that tellurium was mainly bound with organic matter and sulphides (F3A and B), as well as strongly demobilized in the residual fraction. Strong demobilization of tellurium in soils explains the low extraction efficiency of this element with the use of various extractants and the necessity to use ultrasound assistance during tellurium extraction from soils. The results showed that the best extraction efficiency was obtained by using 100 mM citric acid. The use of this extractant (with a low pH) allowed for the extraction of tellurium associated with the F2 fraction (metal forms associated with iron and manganese oxides). The remaining extractants, such as Na_2_EDTA or ammonium tartarate, had a higher pH and under these conditions the tellurium was strongly demobilized. Sequential chemical extraction of certified reference material (CRM 73324) showed that tellurium in this soil was mainly associated with the F3 and R fractions. The sum of the tellurium content in the F0 and F1 (mobile) fractions accounted for about 5% share, while the chemical extraction of CRM with citric acid (during 4 h), showed 10% share of this element extracted from the soil. With the use of citric acid, more tellurium is extracted than what results from the sequential chemical extraction. This is also confirmed by the results of research on real soil samples. In the case of the 58 Edel, 65 Edel or 97 Edel samples, the share of tellurium in the F0 and F1 fractions was higher, and the extraction with citric acid of these soils showed a higher extraction efficiency.

#### 2.1.7. The Matrix Interferences

Soil type can affect the performance of the method. Soil components, such as organic matter and minerals (clay, sand, silt) in varying proportions, affect the physical, mechanical, chemical and water properties of soils. Matrix effects mainly in recovery, and increase or decrease the response during the analysis due to the presence of substances interfering with the detection of the target compounds. In general the effects of soil parameter on determination of tellurium species are poorly published. The research of Salvia et al. [36] based on the ANCOVA model shows that among other things organic carbon–being one of the main component of SOM (soil organic matter) had a significant impact on recovery. This is also evident in our research. Samples of higher organic matter content showed worse extraction efficiency. While it has been also reported that carbon- based compounds does not cause significant polyatomic interferences at masses ^126^Te and ^128^Te [37]. The research of Casiot et al. [38] on soils extract samples exposed to fulvic acid showed that organic matter was not expected to interfere with the species of tellurium in the extract. In our research due to minimalize spectral interferences coming from presence of Ba and Xe ^125^Te, ^126^Te, ^128^Te, ^130^Te isotopes were measured using correction equations (−0.003404 × Xe129) for ^126^Te, (−0.072617 × Xe129) for ^128^Te, and (−0.009437 × Ba137 − 0.154312 × Xe129) for ^130^Te. Toward compensate for matrix effects and drift in analysis of total concentration of elements, a technique with an internal standard was used. Standards, blanks and samples were measured using ^103^Rh as internal standard. Solution of 10 μg/L Rh was moved into all solutions and samples on line, by teeing in tubing on peristaltic pump. The average concentration of Cl^−^ and CO_3_^2−^ in soil is 0.10 g kg^−1^ and about 0.5%, respectively [39]. Because of the large content of chloride and carbonate ions in soil as well as our earlier experience in speciation analysis study, it was decided to check the influence of Cl^−^ and CO_3_^2−^ ions. The matrix effect were tested using NaCl solution (effect of chlorides on separation) and sodium carbonate solution (effect of carbon addition on separation). The chloride interferences were checked by spiking real samples of soil extracts by 10 μg/L of sodium chloride solution. The same was done in the case of carbonate ions. No effect of chlorides and carbonates on the tellurium speciation analysis was observed.

#### 2.1.8. Quality Control of the Speciation Analysis

Limit of detection of tellurium species was determined through measuring a series of standard solutions for two tellurium speciacion forms. The linear model of the concentration dependence on total analyte counts was selected. Using the numerous determinations of the calibration curves, they also helped to calculate the limit of detection (LOD) for each tellurium form. The LOD calculation was based on the following dependence:LOD = (3.3 × S)/b(1)
where: LOD-limit of detection, S-standard deviation value, b-the slope of the calibration curve.

The standard deviation values were determined as a standard deviation for seven sample solutions spiked with known amount of each tellurium species solutions. Limit of detection tellurium species Te(VI) and Te(VI) was 0.002 mg/kg and 0.004 mg/kg, respectively. The limits of quantification (LOQ) were expressed as three times the limit of detection value.

Due to the lack of certified reference materials containing both tellurium forms, the methodology for determining tellurium species was checked on the basis of certified reference material, which was extracted like the real soil samples. The obtained results (Table 2) showed that the CRM sample contain mainly a reduced tellurium form, and its recovery was 105% (the concentration of tellurium from the certificate is 0.4 ± 0.1 mg/kg). Moreover, speciation analysis of the real soil extract (sample 105 Edel) with the addition of 5 μg/L of the mixture of Te(VI) and Te(IV) standards was performed. The results showed tellurium content recovery of 111% for Te(VI) and 80% for Te(IV).

### 2.2. Total Tellurium Concentration

The prepared and validated methodology allowed for the analysis of several soil samples taken from the areas affected by the WEEE processing and sorting plant. The tested soil samples contained small amounts of tellurium. Table 2 presents the concentration of tellurium in studied soils. The total content of tellurium in the samples ranged from 0.02 mg/kg to 0.42 mg/kg. Thanks to the use of a sensitive analytical method such as ICP-MS, it was possible to quantify such small amounts of tellurium. Sequential chemical extraction (Figure 5) showed that in these soils, tellurium was mainly associated with sulphides and organic matter, and with the silicates (residual fraction). These results confirmed our earlier assumptions. The soils in these areas had a forest character; the extracts were dark in color and contained a lot of humic substances. Sequential chemical extraction (Figure 5) made it possible to explain why the efficiency of tellurium extraction with the use of extractants intended for the speciation analysis was characterized by low efficiency. In the studied soils, tellurium was associated with sulphides and silicates; hence, the extraction of this element with extractants such as citric acid, ammonium tartarate, or Na_2_EDTA was characterized by low efficiency.

As reported in the literature, soils tend to be enriched in Te (average around 35 µg/kg), locally near Te deposits [40]. Volcanoes are one source of environmental Te, where volatile forms of Te (H_2_Te) are released through eruptions or hydrothermal activity. For example, Te occurs at 10–1000 mg/kg Te levels in volcanogenic sulfur [41]. While the content of tellurium in soils in the areas associated with the mining of deposits related to Sb-As-Tl ranged from 0.01 to 0.24 mg/kg [42]. Other authors reported that elevated levels of tellurium (maximum 11 mg/kg) are reported in topsoils (<5 cm) around a long-established nickel refinery at Clydach in the Lower Swansea Valley, UK [43]. The tellurium concentration in Japanese soil ranged from 16.2 to 212 ng/g or 0.016 to 0.212 mg/kg [44]. The content of this element in the soil of the Dashuigou tellurium deposit (Sichuan Province, China) was 1.202 mg/kg [45]. Our preliminary results indicate the influence of sorting and processing electrowaste plant on the increasing tellurium concentration in the topsoil of the surrounding areas. As shown in Figure 6, an increased concentration of tellurium occurred in soil samples located north of the electrowaste processing and sorting plant, which is consistent with the prevailing wind direction in this area (north, north-east).

### 2.3. Tellurium Speciation

The study of basic physicochemical parameters (Table 2) showed that the soils in the studied area are slightly acidic. The pH of soil measured in the water was in the range of 3.35–3.85, while the pH of soil measured in KCl was a minimum of 3.11 to a maximum of 3.68. The redox potential (Eh) of the soils in this area was in the range of 349.3–425.5 mV. In the case of tellurium, the +4 and +6 oxidation states have comparable stability. Te(IV) and Te(VI) forms are stable an coexist in pH 4.0 and charge E^0^(Te(VI)/Te(IV)) = +1.02 V [17]. The transformations of many soil components and the introduced pollutants are related to redox transformations, which are directly conditioned by the presence of electron acceptors [46]. By determining the redox potential in soil, it is possible to determine its redox state. The ability of the soil to maintain the redox potential Eh at a sufficiently high level is a measure of the soil resistance to reduction processes. The value of +300 mV proposed by Mortimer is assumed for the boundary between the oxidation and soil reduction conditions. Thus, the measure of soil resistance is the time when the potential drops below the adopted value [47]. The limiting value of redox potential is +300 mV, which corresponds to the reduction of iron (III) to (II) [48]. On this basis, it can be concluded that the soils surrounding the electrowaste treatment plant are characterized by a higher Eh (above 300 mV) and are oxygenated. Therefore, it is not surprising that the results of the speciation analysis showed that the oxidized tellurium form Te(VI) appeared in soil samples. The results presented in Table 2 showed that, in the tested soils, the concentration of both tellurium forms was comparable or that the concentration of the reduced form of tellurium quantitatively exceeded the content of Te(VI). Preliminary studies have shown that the problem is complex and requires in-depth research in this area.

Tellurium is usually present at very low concentrations in environmental samples, which makes it a challenging element to measure in complex matrices. Most tellurium compounds are considered toxic. Except from samples collected close to sources of contamination, tellurium concentrations in soils and sediments are normally at the low ppb (μg/g) level [5]. Te is generally bound by sorption onto clay-sized soil particles rather than in minerals [49]. In particular, a strong association has been observed between Fe^3+^ oxide minerals and tellurium. Te(VI) can be incorporated into the structures of Fe^3+^ oxides, whereas Te(IV) tends to be bound more weakly to Fe^3+^oxides by surface interactions only [4]. However, as our research has shown (Figure 5), in the case of soil samples tested in the area surrounding the WEEE plant, tellurium was mostly associated with F3 (metal related to organic matter and sulphides) and R (residual) fraction. The share of tellurium bound to iron oxides in these soils did not exceed 10%.

## 3. Material and Methods

### 3.1. Sampling Area and Soil Preparation

Samples for this study were taken from the area surrounding the WEEE processing plant, which was located in the Metropolis of Upper Silesia and Zagłębie in Poland. This area is heavily urbanized and is the most industrialized area of Poland. The company collects used devices and components from electrical and electronic devices. Waste is segregated and collected according to the type and stored in containers and big bags on a hard surface. The storage site is covered, which protects the waste from contact with rainfall and prevents the leakage of metal compounds and hazardous substances directly into the soil. In one of the closed halls, manual disassembly of electronic elements and components is carried out, and in another closed hall, the process of “skinning” copper cables is carried out with the use of a cable recycling machine. Soil sampling was performed using a Humax soil sampler to collect undisturbed 30 cm soil cores (two from each place). Soil samples were separated and subjected to chemical analysis after air drying, averaging, and sieving through a sieve with a mesh size of 0.2 µm. The basic physicochemical parameters of soils, such as pH and Eh (redox potential), were collected at fourteen points after the samples were delivered to the laboratory. Eh was measured using the ERPt-111 electrode (Elmetron, Zabrze, Poland) and pH was measured using the ERH-111 electrode (Elmetron, Poland) [50]. Figure 1 presents spatial distribution of the sampling points in the area surrounding the WEEE processing plant. Table 3 shows the geographic coordinates of the soil sampling points. Figure 6 was prepared using Surfer 8 program on the basis of the samples chosen for Te speciation analysis and a total content as well (in the number of 14). On the map locations of analyzed soil samples were marked. In this area a total of 30 soil cores were collected, from which 66 soil samples were then subjected to geochemical analyzes (non-published data).

### 3.2. Apparatus

The total tellurium content was determined using the Elan 6100 DRC-e ICP-MS spectrometer (Perkin Elmer, Waltham, MA, USA). The ICP-MS apparatus was equipped with a standard ICP quartz torch, cross-flow nebulizer, and nickel cones. Table 1 presents operating parameters of the spectrometer. To separate the Te(IV) and Te(VI) species, a speciation apparatus set was applied. It consisted of an HPLC chromatograph (Perkin Elmer, USA) equipped with a Series 200LC Peltier oven, Series 200LC autosampler, and Series 200LC gradient pump. The sample from chromatographic column was introduced to ICP-MS by tubing system, automatic diverter and peristaltic pump. The diverter operates as an automatic switching valve to divert undesired portions of the eluate from the HPLC system to waste before the sample enters the ICP-MS. Soil samples were digested in a microwave oven (Microwave 3000, Anton Paar, Austria). Soil extractions were carried out using an ultrasonic cleaner (Sonic 5, Polsonic, Warszawa, Poland), and then the samples were centrifuged using a Beckman Coulter Avanti JXN-26 centrifuge (20,000 rpm, JA-25.50 Fixed-Angle Aluminum Rotor type).

### 3.3. Reagents

The following substances were used for total tellurium content by ICP-MS analysis: tellurium standard for ICP (Sigma-Aldrich, Buchs, Switzerland), Mix 1 (Sigma-Aldrich, Switzerland), and Rhodium standard (Merck, Darmstadt, Germany). Working standard solutions were obtained by appropriate dilution of the stock standard solutions using acidified (suprapur 65% nitric acid, Merck, Darmstadt, Germany) Milli-Q-Gradient ultra-pure deionized water (Millipore, Milli-Q-Gradient ZMQ5V001). Standard stock solution for HPLC speciation of tellurium (IV) (1 g Te L^−1^) and tellurium (VI) (100 mg Te L^−1^) were prepared by dissolving the appropriate amounts of sodium tellurite (Na_2_TeO_3_) (Aldrich, St. Louis, MO, USA) and potassium tellurate hydrate (K_2_TeO_4_ × H_2_O) (Aldrich, USA) in 0.5% suprapur nitric acid (Merck, Darmstadt, Germany). Moreover 30% spectrally pure hydrochloric acid (Merck, Germany) was used for reduction of Te(IV) standard. For mobile phase preparation the ethylenediaminetetraacetic acid disodium salt dehydrate (99.0–101.0%, Sigma-Aldrich, USA) and potassium hydrogen phthalate (KHP) (≥99.95%, Sigma-Aldrich) were dissolved in water. To check the matrix effects sodium chloride (≥99,0%, Sigma Aldrich, USA) and sodium carbonate (≥99,0%, Sigma Aldrich, USA) were used. Citric acid (≥99.5%, Sigma Aldrich, USA), ammonium tartrate dibasic (≥99.0%, Fluka, Munich, Germany), ethylenediaminetetraacetic acid disodium salt dehydrate (99.0–101.0%, Sigma-Aldrich, USA) were used throughout the extraction studies. 50 mM citric acid (≥99.5%, Sigma Aldrich, USA) was used for diluting Te(IV) and Te(VI) working solutions. Listed compounds were of analytical grade or higher purity.

### 3.4. Sequential Chemical Extraction of Soil

The BCR (the Institute for Reference Materials and Measurements) sequential chemical extraction helped to determine the tellurium forms in soil and the way in which they were bound. According to this procedure, it is possible to isolate metals associated with the distilled water, 3 main fractions, and an additional residual fraction [51]. Soil samples for BCR analysis were selected based on the quantitative results of tellurium analysis in soils. Soil samples from the top layer with the highest Te content were tested. Conditions for the BCR sequential chemical extraction of tellurium are presented in Table 4. The extraction steps were as follows: Fraction F0 (metal dissolved in pore water); Fraction F1 (metal ion exchange and carbonate, acid soluble, mobile); Fraction F2 (reducible, relates to metal forms associated with iron and manganese oxides, mobile); Fraction F3 (F3A and B) (oxidation, relates to metal forms related to organic matter and sulphides, immobile); Fraction R (residual fraction, refers to the metal content in the residue after previous extractions, immobile).

### 3.5. Determination of the Total Tellurium and Tellurium Species Content

0.2 g of soil samples for the total elements analysis were digested in a microwave oven with 5 mL of HNO_3_ (spectral purity, Merck, Germany), 2 mL of H_2_O_2_, and 3 mL of HF (spectral purity, Merck, Germany). The digestion program was: 1400 W, 35 min. After microwave digestion, samples were diluted to 50 mL in polypropylene flasks. Afterwards, they were stored in a fridge at 2–5 ^0^C. Each sample was measured three times using ICP-MS. Samples and standards were delivered with a peristaltic pump. The spectrometer was optimized daily with a 10 µg/L solution (Mg, Cu, Rh, Cd, In, Ba, Ce, Pb, and U) in 1% HNO_3_ Elan 6100 Setup/Stab./Masscal. Solution (Perkin Elmer). The concentrations of tellurium were measured with the internal ^103^Rh standard. Direct determination of tellurium in environmental samples by ICP-MS is often complicated by its low abundance, poor analytical sensitivity, and the presence of Xenon interferences. Therefore ^125^Te, ^126^Te, ^128^Te, ^130^Te isotopes were measured using correction equations (−0.003404 × Xe129) for ^126^Te, (−0.072617 × Xe129) for ^128^Te, and (−0.009437 × Ba137 − 0.154312 × Xe129) for ^130^Te. The best results were obtained with the ^126^Te isotope, similar to the work by Filella & Rodushkin [52]. Standard solutions with Te concentrations from 1 to 25 µg L^−1^ were analyzed to construct a calibration curve with a correlation coefficient of 0.9997. The matrix effect were tested using NaCl solution (effect of chlorides on separation) and sodium carbonate solution (effect of carbon addition on separation). No effect of chlorides and carbonates on the tellurium speciation analysis was observed. The relevant information was written in the text. Operating parameters of the ICP-MS spectrometer are shown in Table 5. Optimal separation conditions are presented in Table 1. The most optimal separation conditions were obtained using 10 mM Na_2_EDTA together with 6 mM KHP at pH in the range of 4.29–4.32 as eluent.

#### 3.5.1. Quality Control of Total Tellurium Concentration

Limits detection of the method of total analysis (LOD) was determined as three times the standard deviation value for the test sample witch were seven spiked samples (5–10 × noise) taken through all the sample preparation steps before and analyzed. The limits of quantification (LOQ) were expressed as three times the limit of detection value. The limit of detection and quantitation of tellurium was 0.006 mg/kg and 0.02 mg/kg, respectively. In order to validate the methodology of the total tellurium content, the certified reference material NCS DC 73324 (China National Analysis Center for Tron and Steel, Beijing, China) was used. The digested CRM sample was analyzed for tellurium content, yielding 105% recovery of tellurium.

#### 3.5.2. Soil Extraction for IC-ICP-MS Analysis

Soil samples subjected to speciation analysis were extracted. Extraction of soil samples is not as efficient as microwave digestion, but high temperature and reagents may change the degree of oxidation element species. For this reason, the method of soil extraction was optimized using Certified Reference Material of soil NCS DC 73324 and NCS DC73322 (China National Analysis Center for Tron and Steel, Beijing, China). The development of the extraction methodology included the use of extraction support by shaking and ultrasound. The extraction efficiency of soils for the tellurium species content was checked by shaking 1 g of the Certified Soil Reference Material (NCS DC73324, NCS DC73322) for 2 h on a shaker (145 rpm) using 10 mL of extractants, such as 100 mM citric acid, 20 mM Na_2_EDTA, 300 mM ammonium tartrate, and water. The degree of tellurium extraction from soils was tested by checking the effect of extraction time and the type of extracting agent at a temperature of 30 °C using an ultrasonic washer. The effect of extraction time on the degree of tellurium leaching after 1, 2, 3, and 4 h was investigated. A gradual extraction of the soils was used, in which a fresh extracting agent (10 mL) was added and, after 1 h extraction, the sample was centrifuged and fresh extractant was added. For this purpose, 1 g of a portion of the Certified Reference Material (NCS DC73324) was extracted with various eluents (100 mM citric acid, 20 mM Na_2_EDTA, 300 mM ammonium tartrate, and a mixture of 100 mM citric acid with 20 mM Na_2_EDTA pH = 3.8), and then total tellurium content was determined. After extraction, the soil samples were centrifuged using a Beckman Coulter Avanti JXN-26 centrifuge (20,000 rpm, 8 min, 4 °C), and the supernatant was decanted and filtered through a PES syringe filter with a pore diameter of 0.22 µm.

## 4. Conclusion

The use of combined IC-ICP-MS techniques allowed for the quantification of the tellurium species within 4 min. This required optimization and validation of the extraction methodology, separation conditions, and determination of the tellurium species. The optimized methodology allowed interesting research results to be obtained.

According to the literature, tellurium is a naturally occurring element found in its pure state in the Earth’s crust in the amount of 0.001–0.01 mg/kg [9]. However, our results have shown that the concentration of tellurium in soil can be up to 10 times higher and amount to 0.166 mg/kg, despite the fact that Te(VI) is less thermodynamically stable but more abundant than Te(IV) [18]. Comparable amounts of both tellurium ionic forms were found in soils from the area surrounding the WEEE sorting and processing facility, or the reduced form of this element predominated. The obtained results indicated that research on the influence of the electrowaste processing and sorting plant on the surrounding soil should be continued by testing more soil samples, calculating geochemical coefficients, and examining environmental conditions in greater detail.

The work uses the Surfer 8 program, Microsoft Windows 10 Home edition, Microsoft Office Standard 2016, Elan Version 3.4, Chromera Version 2.0.

## Figures and Tables

**Figure 1 molecules-26-02651-f001:**
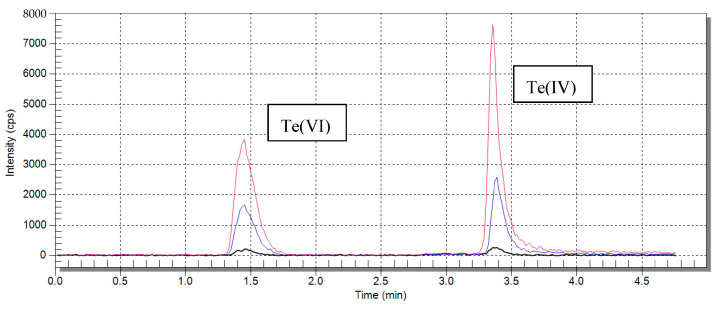
Superimposed chromatograms for tellurium speciation. Calibration curves obtained with measurement of 1 µg/L, 10 µg/L, and 25 µg/L Te(IV) and Te(VI) standard solution.

**Figure 2 molecules-26-02651-f002:**
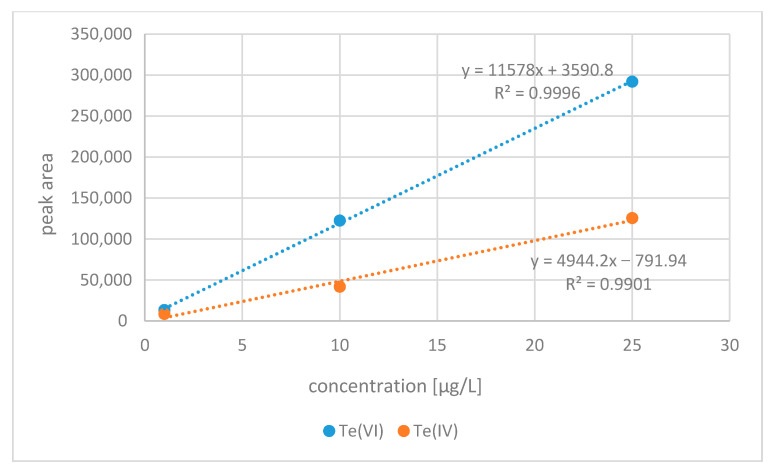
Calibration curves of tellurium species separation.

**Figure 3 molecules-26-02651-f003:**
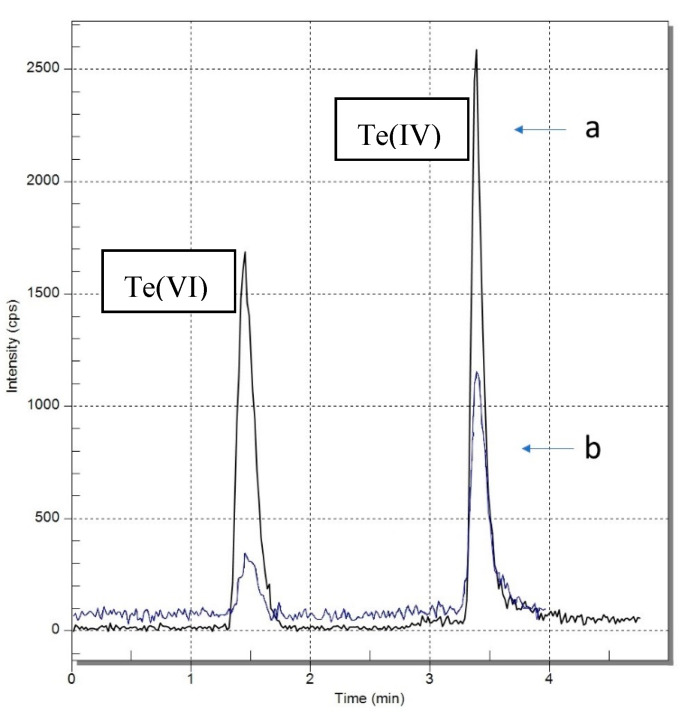
Chromatograms of tellurium species: (a) 10 µg/L Te(IV) and Te(VI) standard solution, (b) real soil samples 70 Edel.

**Figure 4 molecules-26-02651-f004:**
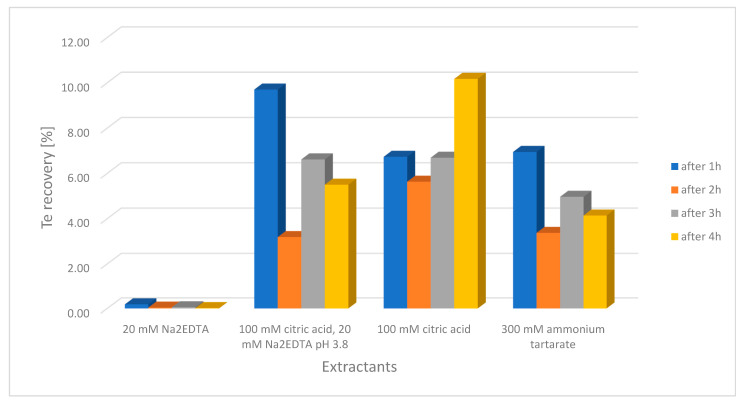
Ultrasound assisted tellurium extraction using various extractants.

**Figure 5 molecules-26-02651-f005:**
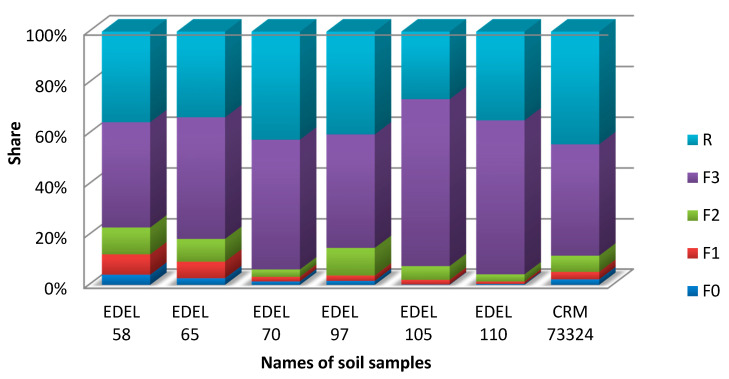
Sequential chemical extraction of tellurium in soil; F0 fraction (metal dissolved in pore water), F1 fraction (metal ion exchange and carbonate); F2 fraction (metal forms associated with iron and manganese oxides), F3 fraction (F3A and B) (metal related to organic matter and sulphides), R fraction (residual fraction).

**Figure 6 molecules-26-02651-f006:**
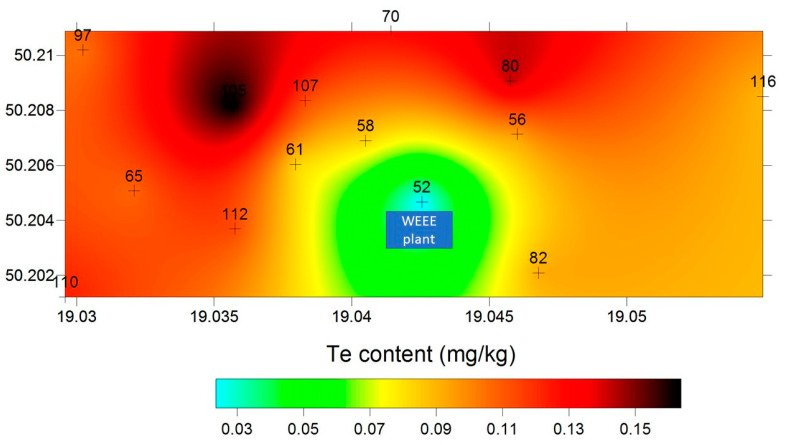
Spatial distribution of the total tellurium content in soil samples in the area surrounding the WEEE processing plant. The map was prepared using Surfer 8 software on the basis of 14 soil samples chosen for Te speciation analysis.

**Table 1 molecules-26-02651-t001:** Chromatographic conditions.

Parameter	Value
Tellurium	
Separation column	Hamilton PRP-X100 4.6 mm × 150 mm, 5 µm
Temperature	30
Mobile phase	10 mM Na_2_EDTA, 6 mM KHP pH = 4.29–4.32
Elution program	4 min
Retention time of Te species [min]	Te(VI)-1.46 Te(IV)-3.18
Flow rate during analysis [mL/min]	1.2
Flow rate during the rinsing [mL/min]	1.2
Volume of sample [μL]	200

**Table 2 molecules-26-02651-t002:** Concentration of tellurium and its species in soil.

Sample	Total Tellurium Concentration after Digestion [mg/kg]	Total Tellurium Concentration after Extraction [mg/kg]	Extraction Efficiency [%]	Te(VI) Concentration [mg/kg]	Te(IV) Concentration [mg/kg]	Sum Te(IV) and Te(VI)	pH Value in H_2_O	pH Value in KCl	Eh [mV]
CRM	0.42	0.043	10	0.004	0.036	0.040	3.46	3.12	362.2
52 Edel	0.021	0.010	47	0.005	0.009	0.013	3.54	3.22	387.8
56 Edel	0.095	0.022	24	0.010	0.006	0.016	3.82	3.54	395.2
58 Edel	0.083	0.025	30	0.013	0.009	0.022	3.83	3.49	349.3
61 Edel	0.083	0.028	33	0.011	0.011	0.022	3.46	3.12	412.8
65 Edel	0.108	0.029	27	0.020	0.007	0.027	3.85	3.59	384.5
70 Edel	0.129	0.016	12	0.003	0.009	0.013	3.43	3.14	425.5
80 Edel	0.143	0.016	11	0.003	0.005	0.008	3.92	3.62	384.7
82 Edel	0.092	0.014	16	0.009	0.008	0.017	4.09	3.58	365.8
97 Edel	0.105	0.022	21	0.017	0.005	0.022	4.33	3.68	376.2
105 Edel	0.166	0.017	10	0.004	0.013	0.017	3.62	3.46	375.8
107 Edel	0.114	0.013	11	0.002	0.005	0.007	3.45	3.15	382.4
110 Edel	0.124	0.011	9	0.005	0.011	0.015	3.35	3.11	398.5
112 Edel	0.114	0.014	12	0.002	0.009	0.011	3.69	3.39	396.5
116 Edel	0.090	0.019	21	< LOD	0.017	0.017	3.42	3.14	411.1

**Table 3 molecules-26-02651-t003:** Geographic coordinates of the soil sampling points.

Sample No.	N Latitude	E Longitude
52 Edel	50,204670	19,042550
56 Edel	50,207130	19,046020
58 Edel	50,206900	19,040510
61 Edel	50,206040	19,037950
65 Edel	50,205070	19,032090
70 Edel	50,210880	19,041430
80 Edel	50,209070	19,045770
82 Edel	50,202090	19,046790
97 Edel	50,210190	19,030230
105 Edel	50,208170	19,035690
107 Edel	50,208350	19,038310
110 Edel	50,201210	19,029580
112 Edel	50,203690	19,035750
116 Edel	50,208510	19,054950

**Table 4 molecules-26-02651-t004:** Conditions for the BCR sequential chemical extraction of tellurium.

Extraction Rate	Form	Extracting Reagent
0	Dissolved in pore water	distilled water
F1	Ion exchange and carbonate	20 mL 0.11 M CH_3_COOH 16 h, continuous mixing
F2	Oxide	20 mL 0.1 M NH_2_OH.HCl (pH 2, supplied HNO_3_) 16 h, continuous mixing
F3	Organic	(A) 10 mL H_2_O_2_ 30% pH 2 2 h water bath 85 ± 2 °C 10 mL H_2_O_2_ 8.8 M pH = 2 2 h water bath 85 ± 2 °C (B) 25 mL 1 M NH_4_OAc pH 2 16 h continuous mixing (pH 2, supplied HNO_3_)
R	Residual	6 mL HCl, 2 mL HNO_3_, 3 mL HF microwave digestion Anton Paar Microwave 3000, power 1400 W, time 45 min.

**Table 5 molecules-26-02651-t005:** Operating parameters of the ICP-MS spectrometer.

Parameter	Value
ICP-MS	
RF power [W]	1125
Plasma gas flow [L/min]	15
Nebulizer gas flow [L/min]	0.76–0.82
Auxiliary gas flow [L/min]	1.15–1.16
Nebulizer type	Cross flow
Plasma torch	Quartz
Scanning mode	Peak hopping
Dwell time [ms]	100
Sweeps/reading	20
Number of replicates	3

## Data Availability

Not applicable.

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
