# Peer review of "Development of a Tellurium Speciation Study Using IC-ICP-MS on Soil Samples Taken from an Area Associated with the Storage, Processing, and Recovery of Electrowaste"

_molecules, 2021, doi:10.3390/molecules26092651_

Round 1

Reviewer 1 Report

The topic of the article, tellurium speciation in soil, is very interesting and could be of interest to environmental scientists and the broader public. 

However, the results are not accurately presented and discussed and the article needs to be revised before publishing.

Lines 77-88: The paragraph is very confusing. You have two opposing sentences: It is difficult to detect tellurium with ICP-MS because of its poor sensitivity to tellurium due to the high ionization energy and the fact that it has many isotopes, nearly all interfered by isobaric Xe.   Fortunately, accurate results can be obtained in the majority of matrices using both the 125Te and 126Te isotopes.

You need to rewrite these sentences and explain why is using 125Te and 126Te isotopes and how are problems such as high ionization and Xe interferences avoided.

Paragraph 2.4 Why did you perform sequential chemical extraction? And why you did not use these results in the discussion? As I can see from Figure 1. most of the Te is bounded to organic matter and sulfides or in residual fraction and you have small concentrations of Te in F0 and F2. You need to correlate these results with the results obtained with citric acid extraction.

Paragraph 2.5. The correction equations. How did you calculate it?

Paragraph 2.5.1 LOD described in this paragraph is the detection limit of the instrument. I did not find any data in your article on the LOD of the whole developed method and I believe that this is the major deficiency of the article.  If you are validating a new method, you need to provide the method detection limit expressed as mg/kg of Te and the method accuracy expressed as RSD. You do not use a blank sample for MDL measurements but a spiked sample with the spike level 5-10 x noise.

Paragraph 3.1. This paragraph describes only eluation optimization so you should change the title.

Paragraphs 3.1.1. and 3.1.2. I would switch places of these paragraphs so the first one should be explaining Te separation and the next one explaining chromatographic separation.

Figure 3. There is no calibration curve presented

Why did you use CRM for citric acid extraction optimization when it is certified only for total Te? You should perform BCR sequential extraction on CRM soil and then try to correlate the results of Te concentrations in F0 and F1 with the results of citric acid extraction.

Author Response

Response to editor’s and reviewers’ comments

Manuscript ID.: molecules-1170128

Title: Development of a Tellurium Speciation Study Using IC-ICP-MS on Soil Samples Taken from an Area Associated with the Storage, Processing, and Recovery of Electrowaste

Journal: Molecules

Article Type: Original research paper

# Reviewer comments in black

# Response in blue

Response to editor

Dear Ms. Jovana Curic, MSc, Assistant Editor, MDPI

We are submitting a revised manuscript (Molecules). We thank the editor and the reviewers for their suggestions. We are grateful for the helpful comments and advices from editors and reviewers. It is a valuable chance to improve our manuscript.

We have adjusted the manuscript and have provided a response for each of the comments and noted the changes made. We believe the new manuscript is more logic streamlined than the old one. For each change in the manuscript, the modified sentence(s) along with line numbers of changes in the revised version are provided, so that you could be easily identified. Major revision included the change of title, tables, figures, text, references. The changes are marked in manuscript (track).

We sincerely hope that our revised manuscript would meet the standards of publication.

Yours sincerely,

dr hab. Magdalena Jabłońska-Czapla, Prof. IPIS PAN

PhD

Institute of Environmental Engineering of Polish Academy of Science
Sklodowska-Curie 34 Street, 41-819 Zabrze, POLAND
tel. +48 32 2716481, fax. +48 32 2717470
e-mail: [email protected]

Reviewer #1:

Lines 77-88: The paragraph is very confusing. You have two opposing sentences: It is difficult to detect tellurium with ICP-MS because of its poor sensitivity to tellurium due to the high ionization energy and the fact that it has many isotopes, nearly all interfered by isobaric Xe.   Fortunately, accurate results can be obtained in the majority of matrices using both the 125Te and 126Te isotopes.

You need to rewrite these sentences and explain why is using 125Te and 126Te isotopes and how are problems such as high ionization and Xe interferences avoided.

Thank you very much for this remark.

These sentences were contradictory and have been corrected.

“Direct determination of tellurium in geological samples by inductively coupled plasma mass spectrometry (ICP-MS) is often complicated by its low abundance, poor analytical sensitivity, and the presence of xenon interferences. Fortunately, accurate results, with low xenon interferences, can be obtained in the majority of matrices using both the 125Te and 126Te isotopes.”

Paragraph 2.4 Why did you perform sequential chemical extraction? And why you did not use these results in the discussion? As I can see from Figure 1. most of the Te is bounded to organic matter and sulfides or in residual fraction and you have small concentrations of Te in F0 and F2. You need to correlate these results with the results obtained with citric acid extraction.

Thank you very much for this remark.

I performed sequential chemical extraction to find out what fraction is mainly related to tellurium in the soils under study.

The discussion was completed and the results of sequential chemical extraction were referred to.  In a new point 3.1.5 Sequential chemical extraction

Paragraph 2.5. The correction equations. How did you calculate it?

Calibration curve (y=ax+b) were obtained with measurement of 1 μg/L, 10 μg/L, and 25 μg/L standard solutions for Te. A linear model of the dependence of concentration of the total number of analyte counts was selected. The coefficient of determination of calibration curves R2 was between 0.9997.

Paragraph 2.5.1 LOD described in this paragraph is the detection limit of the instrument. I did not find any data in your article on the LOD of the whole developed method and I believe that this is the major deficiency of the article.  If you are validating a new method, you need to provide the method detection limit expressed as mg/kg of Te and the method accuracy expressed as RSD. You do not use a blank sample for MDL measurements but a spiked sample with the spike level 5-10 x noise.

Method detection limit (MDL) was calculated to determine analyte detection limit in soil samples. Test solutions for checking MDL were seven spiked samples (5-10 x noise) taken  through all the sample preparation steps before and analyzed. The method limit of detection was calculated in the following dependence:

MDL=3.3S

where: MDL – method detection limit, S- replicate samples analysis standard deviation value.

The limit of quantification (LOQ)  was expressed as:

LOQ=3LOD

where: LOQ – method limit of quantification, LOD-  detection limit of the method value

The method limit of detection and quantitation of tellurium was 0.006 mg/kg and 0.02 mg/kg respectively. In order to validate the methodology of the total tellurium content, the certified reference material NCS DC 73324 (China National Analysis Center for Tron and Steel, Bei-jing, China) was used. The digested CRM sample was analyzed for tellurium content, yielding 105 % recovery of tellurium.

Paragraph 3.1. This paragraph describes only eluation optimization so you should change the title.

Thank you very much for this remark.

Paragraph 3.1 it concerns tellurium speciation. The following subsections describe the optimization steps. I have added a subsection 3.1.1 Elution optimization.

Paragraphs 3.1.1. and 3.1.2. I would switch places of these paragraphs so the first one should be explaining Te separation and the next one explaining chromatographic separation.

Thank you very much for this remark.

The paragraphs have been replaced in accordance with the reviewer's remark.

Figure 3. There is no calibration curve presented

Something must have been damaged while applying the article. I have uploaded a new chart.

Why did you use CRM for citric acid extraction optimization when it is certified only for total Te? You should perform BCR sequential extraction on CRM soil and then try to correlate the results of Te concentrations in F0 and F1 with the results of citric acid extraction.

Thank you very much for this remark. Of course, we did a sequential chemical extraction of the certified reference material, these results were not just posted. Due to the comment of the reviewer, Figure 4, which presents the results of the sequential chemical extraction of CRM 73324, has been changed. Moreover, the results of BCR extraction of certified material have been discussed (3.1.5 Sequential chemical extraction) and compared with those obtained for CRM during the extraction of this soil with citric acid.

Reviewer 2 Report

The manuscript entitled “Development of a Tellurium Speciation Study Using HPLC-ICP-MS on Soil Samples Taken from an Area Associated with the Storage, Processing, and Recovery of Electrowaste” describes the validation of a method for quantifiying Te in soil. The method enables the authors to differentiate between two species of Te: tellurite and tellurate. The method consists in an extraction step targeting the soluble or exchangeable fraction, a separation step using ion chromatography to separate the two forms of Te, and the detection using ICP-MS. Such methods allowing to quantify specific species in soil are relatively rare and this study provides a good basis for a procedure which could be optimized for other soils. Therefore, I think this study is interesting and should be published.

However, there are several important points to improve as listed in detail below. At first, important information have to added in the methods, second, in this current form, the discussion is too limited. It would be very important for the understanding of the method to discuss the speciation of Te in soils in more details, on one side, and to add data on other types of soils in order to evaluate the general applicability of the method. Indeed, the authors provide data from many samples but all of them from the same area without explaining what differentiate these samples from each other. This has the effect that the discussion does not bring any new understanding about the speciation of Te in this particular area. Furthermore, since no other soils were tested, one is also not able to assess from the manuscript how robust the method is.

Therefore, I recommend major revisions for this manuscript taking into account the comments listed below.

General:

I recommend using the term “ion chromatography” (IC) instead of HPLC. Indeed, HPLC is referring more to the system (pump, column) than to the separation mechanism. When one carries out IC, the use of a HPLC system or similar is implied. Therefore, I think IC-ICP-MS is a more accurate and informative abbreviation for the proposed method because it is then clear that one uses ion chromatography as a separation mechanism prior to ICP-MS, and it is shorter.

Introduction:

Line 35: Add “(VI)” after “tellurate”.

Line 68: “best solution” Explain what are the criteria or requirements which support this statement. For instance, one argument could be that a simultaneous determination of the two species allows to calculate a Te mass balance if the total content is measured separately. This is not possible in the “difference” methods since the total content is used in the calculation of one of the species. Therefore, the presence of other species (e.g. colloidal form) cannot be detected using such methods.

The solid-phase extraction methods should also be discussed in the introduction since they are an important alternative in cases where the concentrations are very low (enrichment of the analyte). Here are some references which could be added:

Online simultaneous speciation of ultra-trace inorganic antimony and tellurium in environmental water by polymer monolithic capillary microextraction combined with inductively coupled plasma mass spectrometry

https://doi.org/10.1016/j.sab.2020.105854

Simultaneous speciation of inorganic selenium and tellurium by inductively coupled plasma mass spectrometry following selective solid-phase extraction separation

DOI: 10.1039/b310318h

Speciation analysis of tellurium by solid-phase extraction in the presence of ammonium pyrrolidine dithiocarbamate and inductively coupled plasma mass spectrometry

DOI: 10.1007/s00216-003-1895-0

Even if it was not applied for soil, the method described in the following paper should also be compared to the proposed method since it seems very similar:

Determination of selenium and tellurium compounds in biological samples by ion chromatography dynamic reaction cell inductively coupled plasma mass spectrometry

DOI: 10.1016/j.chroma.2007.12.065

Line 84: Give an order of magnitude for the LODs reported in the literature for such systems.

In addition, I strongly recommend to add explanations about Te-chemistry in the introduction. In particular, it is very important to explain the possible species which can exist in soils for naturally occurring and anthropogenic Te. This would allow a better understanding on the functioning of the method.

Material and methods:

Line 106: the detailed position of the sampling location should be given with an explanation of the sample labels (EDEL-XXX). Furthermore, the expected differences in terms of Te-content should be described in detail. Which ones are the control samples? Which samples are expected to be more contaminated? What was the natural background in this area? Was pollution observed or was it just assumed?

Line 108: please replace “diameter” with “mesh size” otherwise it refers to the diameter of the sieve itself.

Line 120: explain how the systems were hyphenated (flow splitting, etc.)

Line 123: Provide the g-force or rpm+rotor type for the centrifuge step, otherwise, it is not reproducible.

Line 128: “acidified” with which acid?

Line 155: what is the difference between F3-A and F3-B? Were both methods used or only one of them?

The figure 1 and the corresponding results should be discussed in the results and discussion part. Also the title should be removed since the figure is explained in the caption. The y-axis should be named. The x-axis is not informative since the reader doesn’t know the meaning of the labels.

Please check the numbering of tables and figures and their position in the text. It seems they were mixed up.

Line 182: please clarify if you used the intensity values or the concentrations values for calculating the LOD. Usually, one uses three times the standard deviation of the intensity of a blank sample divided by the slope of the calibration curve. Furthermore, the determination of the LOD requires a high certainty on the slope, which is not provided with only three points. I recommend determining the LOD using low concentrations and more standards (e.g. 5 + a blank).

The description of the IC-method is missing (flow rate, eluent compositions and pH-values which were tested, injection volume, etc.). Also the data analysis (software, integration method, etc.) should be described.

In general information about replication is missing.

Resuts and discussion:

Lines 200-202: redundant with the method.

Table 3 belongs to the method part.

For a better understanding of the separation process, it is important to discuss the electric charge of the Te-species at the different tested pH-values and how it influences these processes.

Figure 2: add a chromatogram of a soil sample for comparison. Why is the Te(IV) signal larger than the Te(VI) signal? The concentrations are supposed to be the same.

Figure 3: the data points are not visible on the version I received.

Line 253: Please clarify which extraction (BCR or for the IC-ICP-MS) is discussed.

Lines 253-261: this part belongs to the methods section. The conditions of the sonication and centrifugation steps should be explained in more details the methods section.

Line 273: Was the total Te-content determined in the extract?

Figure 3: Replace “extrahents” with “extractants”. Remove the title. Please replace the concentrations with the recovery values. This will be more informative for the reader. Add standard deviations on the graph.

The matrix effect should be tested using e.g. standard addition with more standards in order to compare the slopes in the matrix and in pure water. Or were the standards matrix matched? In this case this should be mentioned in the method.

Lines 316-318: I disagree with this statement since (from the reader perspective) there are not enough information on the measured samples to discuss any effect of the plant on the soil pollution.

Table 4: Please discuss why the extraction efficiency is varying so strongly between samples (10-47%). Use µg/kg for better readability.

For the Te-speciation I recommend using an Eh-pH diagram for a clearer discussion. It has to be noted that the speciation was considered only from the thermodynamic point of view. Depending on the chemical form in which Te in entering the soil, metastable species could also be formed (e.g. colloids or as complex with SOM).

Lines 347-349: this contradicts the BCR-results where the concentration in the Fe/Mn-oxides phases was quite low.

Author Response

Response to editor’s and reviewers’ comments

Manuscript ID.: molecules-1170128

Title: Development of a Tellurium Speciation Study Using IC-ICP-MS on Soil Samples Taken from an Area Associated with the Storage, Processing, and Recovery of Electrowaste

Journal: Molecules

Article Type: Original research paper

# Reviewer comments in black

# Response in blue

Response to editor

Dear Ms. Jovana Curic, MSc, Assistant Editor, MDPI

We are submitting a revised manuscript (Molecules). We thank the editor and the reviewers for their suggestions. We are grateful for the helpful comments and advices from editors and reviewers. It is a valuable chance to improve our manuscript.

We have adjusted the manuscript and have provided a response for each of the comments and noted the changes made. We believe the new manuscript is more logic streamlined than the old one. For each change in the manuscript, the modified sentence(s) along with line numbers of changes in the revised version are provided, so that you could be easily identified. Major revision included the change of title, tables, figures, text, references. The changes are marked in manuscript (track).

We sincerely hope that our revised manuscript would meet the standards of publication.

Yours sincerely,

dr hab. Magdalena Jabłońska-Czapla, Prof. IPIS PAN

PhD

Institute of Environmental Engineering of Polish Academy of Science
Sklodowska-Curie 34 Street, 41-819 Zabrze, POLAND
tel. +48 32 2716481, fax. +48 32 2717470
e-mail: [email protected]

Reviewer #2:

The manuscript entitled “Development of a Tellurium Speciation Study Using HPLC-ICP-MS on Soil Samples Taken from an Area Associated with the Storage, Processing, and Recovery of Electrowaste” describes the validation of a method for quantifiying Te in soil. The method enables the authors to differentiate between two species of Te: tellurite and tellurate. The method consists in an extraction step targeting the soluble or exchangeable fraction, a separation step using ion chromatography to separate the two forms of Te, and the detection using ICP-MS. Such methods allowing to quantify specific species in soil are relatively rare and this study provides a good basis for a procedure which could be optimized for other soils. Therefore, I think this study is interesting and should be published.

However, there are several important points to improve as listed in detail below. At first, important information have to added in the methods, second, in this current form, the discussion is too limited. It would be very important for the understanding of the method to discuss the speciation of Te in soils in more details, on one side, and to add data on other types of soils in order to evaluate the general applicability of the method. Indeed, the authors provide data from many samples but all of them from the same area without explaining what differentiate these samples from each other. This has the effect that the discussion does not bring any new understanding about the speciation of Te in this particular area. Furthermore, since no other soils were tested, one is also not able to assess from the manuscript how robust the method is.

Therefore, I recommend major revisions for this manuscript taking into account the comments listed below.

General:

I recommend using the term “ion chromatography” (IC) instead of HPLC. Indeed, HPLC is referring more to the system (pump, column) than to the separation mechanism. When one carries out IC, the use of a HPLC system or similar is implied. Therefore, I think IC-ICP-MS is a more accurate and informative abbreviation for the proposed method because it is then clear that one uses ion chromatography as a separation mechanism prior to ICP-MS, and it is shorter.

Thank you very much for this remark.

I agree with the reviewer's remark. In my research I have repeatedly used the abbreviation IC-ICP-MS. In keeping with the remark throughout the text, I will change the acronym HPLC-ICP-MS to IC-ICP-MS.

The title has been change into “Development of a Tellurium Speciation Study Using IC-ICP-MS on Soil Samples Taken from an Area Associated with the Storage, Processing, and Recovery of Electrowaste”.

Introduction:

Line 35: Add “(VI)” after “tellurate”.

I applied a correction and at the moment the sentence is: “Tellurium is considered toxic and teratogenic, and there are indications that tellurite, Te(IV), can be more toxic than tellurate, Te(VI) [3].”

Line 68: “best solution” Explain what are the criteria or requirements which support this statement. For instance, one argument could be that a simultaneous determination of the two species allows to calculate a Te mass balance if the total content is measured separately. This is not possible in the “difference” methods since the total content is used in the calculation of one of the species. Therefore, the presence of other species (e.g. colloidal form) cannot be detected using such methods.

Thank you very much for this remark.

I have made a correction in line 68.

I have corrected this sentence to make it clearer why this type of technique is the "best solution". The current form:

“The best solution is to use techniques that allow simultaneous determination of Te(IV) and Te(VI), which allows for calculate a Te mass balance if the total content is measured sepa-rately.”

The solid-phase extraction methods should also be discussed in the introduction since they are an important alternative in cases where the concentrations are very low (enrichment of the analyte). Here are some references which could be added:

Online simultaneous speciation of ultra-trace inorganic antimony and tellurium in environmental water by polymer monolithic capillary microextraction combined with inductively coupled plasma mass spectrometry

https://doi.org/10.1016/j.sab.2020.105854

Simultaneous speciation of inorganic selenium and tellurium by inductively coupled plasma mass spectrometry following selective solid-phase extraction separation

DOI: 10.1039/b310318h

Speciation analysis of tellurium by solid-phase extraction in the presence of ammonium pyrrolidine dithiocarbamate and inductively coupled plasma mass spectrometry

DOI: 10.1007/s00216-003-1895-0

Thank you very much for this remark.

The literature items proposed by the reviewer were quoted.

In the introduction, I added a text concerning the solid-phase extraction methods:

One of the ways tellurium speciation in environmental samples is the use of the solid-phase extraction methods. They are a particularly important alternative especially in the case of a very low tellurium concentrations [10-12].

[10] X. Ou, C. Wang, M. He, B. Chen, B. Hu, Online simultaneous speciation of ultra-trace inorganic antimony and tellurium in environmental water by polymer monolithic capillary microextraction combined with inductively coupled plasma mass spectrometry. Spectrochim. Acta B 168 (2020) 105854. https://doi.org/10.1016/j.sab.2020.105854

[11] C. Yu, Q. Cai, Z.X. Guo, Z. Yang, S.B. Khoo, Simultaneous speciation of inorganic selenium and tellurium by inductively coupled plasma mass spectrometry following selective solid-phase extraction separation. J. Anal. At. Spectrom. 19 (2004) 410-413.https://doi.org/10.1039/B310318H

[12] C. Yu, Q. Cai, Z.X. Guo, Z. Yang, S.B. Khoo, Speciation analysis of tellurium by solid-phase extraction in the presence of ammonium pyrrolidine dithiocarbamate and inductively coupled plasma mass spectrometry. Anal. Bioanal. Chem. 376 (2003) 236–242. https://doi.org/10.1007/s00216-003-1895-0

Even if it was not applied for soil, the method described in the following paper should also be compared to the proposed method since it seems very similar:

Determination of selenium and tellurium compounds in biological samples by ion chromatography dynamic reaction cell inductively coupled plasma mass spectrometry

DOI: 10.1016/j.chroma.2007.12.065

Thank you very much for this remark. Unfortunately, I cannot agree with reviewer's remark. The technique used in the proposed citation may be similar, that is, it uses IC-ICP-MS, but additionally a DRC chamber is used here. In addition, gradient elution and other eluents are used in this work.

I quoted this article as item [23] in the sentence:

“The application of hyphenated techniques such as HPLC-ICP-MS or IC-ICP-MS allows for speciation analysis and simultaneous determination of several ionic forms of elements [22,23].”

[23] C.Y. Kuo, S.J. Jiang, Determination of selenium and tellurium compounds in biological samples by ion chromatography dynamic reaction cell inductively coupled plasma mass spectrometry. J. Chromatogr. A. 1181 (2008) 60-66. DOI: 10.1016/j.chroma.2007.12.065

Line 84: Give an order of magnitude for the LODs reported in the literature for such systems.

Thank you very much for this remark. I changed my sentence below to add an example of LOD during tellurium determination using hyphenated technique.

“Thanks to this, it is possible to combine analytical techniques in tellurium speciation studies using ICP-MS, with excellent sensitivity and ultratrace amounts of tellurium detection [24-28], for example 0.56 ng/L when analyzing Te(IV) in water [22].”

In addition, I strongly recommend to add explanations about Te-chemistry in the introduction. In particular, it is very important to explain the possible species which can exist in soils for naturally occurring and anthropogenic Te. This would allow a better understanding on the functioning of the method.

Thank you very much for this remark. The introduction has been refined and described about The-chemistry. Additional literature items have been entered.

Tellurium is one of the chalcophile elements that belong to group 16 in the Periodic Table. Its chemical behavior is similar to selenium and in the natural environment can exist in several redox states: telluride (–II), elemental tellurium (0),tellurite (IV) and tellurate (VI) both in organic and inorganic forms. Although chemical properties of tellurium are well known there is few information about its chemical behavior in natural systems. Under ordinary environmental conditions (oxic systems, circumneutral pH, absence of ligands other than those derived from water), the dominant species should be those of TeIV, while TeVI species will be potent oxidants [3]. TeIV/VI ratio can differ under different environmental and biological conditions but studies in natural systems are lacking due to extremely low Te concentrations  in geological, environmental, and biological samples and most of them is focused on developing the research methodology and rather concerns the total content of tellurium. Research of Hai-Bo Qin et al. [4] showed that in abandoned mine tailings contaminated soil Te was present as a mixture of Te(VI) and Te(IV) species, and Fe(III) hydroxides were the host  phases for Te(IV), and Te(VI), but Te(IV) could be also retained by illite. The values for total tellurium in soil and sediments vary in general from less than one ppb to a few ppb (mg/g) depending on locations and sources of contamination [5]. Te content in some types in soils from USA was in the range 0.02 and 0.69 mg/kg [6]. Ferri et al. [7] studied tellurium species concentration in Soil NIST SRM 2709 and showed that in this material the content of inorganic tellurium species was 50% Te(VI) and Te(IV). Harada and Takahashi [8] studied the distribution and speciation of Te between the solid and aqueous phases in synthetic soil. Under oxen conditions Te was mainly associated with iron (III) hydroxides, and Te(IV) and Te(VI) species were both found to inner-sphere complexes.

[3] M. Filella, C. Reimann, M. Biver, I. Rodushkin, K. Rodushkina, Tellurium in the environment: current knowledge and identification of gaps. Environ. Chem. 16 (2019) 215-228 https://doi.org/10.1071/EN18229 ].

[4] H-B. Qin, Y. Takeichi, H. Nitani, Y. Terada, Y. Takahashi, Tellurium Distribution and Speciation in Contaminated Soils from Abandoned Mine Tailings: Comparison with Selenium. Environ. Sci. Technol. 51 (2017) 6027–6035 https://doi.org/10.1021/acs.est.7b00955 ]

[6] K. GOVINDARAJU, 1994 COMPILATION OF WORKING VALUES AND SAMPLE DESCRIPTION FOR 383 GEOSTANDARDS, 18, S1, (1994), 1-158. https://doi.org/10.1046/j.1365-2494.1998.53202081.x-i1

[7] Ferri T, Rossi S, Sangiorgio P (1998). Simultaneous determination of the speciation of selenium and tellurium in geological matrices by use of an iron(III)-modified chelating resin and cathodic

stripping voltammetry. Analytica Chimica Acta 361, 113–123. doi:10.1016/S0003-

2670(98)00021-X

Material and methods:

Line 106: the detailed position of the sampling location should be given with an explanation of the sample labels (EDEL-XXX). Furthermore, the expected differences in terms of Te-content should be described in detail. Which ones are the control samples? Which samples are expected to be more contaminated? What was the natural background in this area? Was pollution observed or was it just assumed?

Thank you very much for this remark. New Figure 1 and Table 1 have been included. Figure 1 shows the spatial distribution of sampling points, and table 1 shows the geographic coordinates of the sampling points. We didn't know which points could be more tellurium contaminated. We suspected that this was the area closest to the WEEE facility. As it turned out, however, the highest concentration of tellurium in soils was found in soil samples from topsoil spaced at a distance from the WEEE plant, the points being located in the direction of the prevailing winds. Soil control samples were collected in the same way as those in the present study in the area not subject to pressure from industry or the WEEE processing plant in Lublin (in eastern Poland). In this article, we focus on presenting the methodology developed for this research. A work is under preparation (being completed) in which we present the results of over 100 soil samples in which critical elements are tested, their spatial distribution, the influence of climatic conditions, correlation between magnetic sustainability and the concentration of critical elements in the soil.

Soil samples were collected at points where the initial magnetic screening showed a higher value of magnetic susceptibility.

Line 108: please replace “diameter” with “mesh size” otherwise it refers to the diameter of the sieve itself.

Thank you very much for this remark. I amended it and now this sentence is:

“Soil samples were separated and subjected to chemical analysis after air drying, averag-ing, and sieving through a sieve with a mesh size of 0.2 µm.”

Line 120: explain how the systems were hyphenated (flow splitting, etc.)

Thank you very much for this remark. The sentence has been corrected and reads as it stands:

“The sample from chromatographic column is  introduced to ICP-MS by tubing system, automatic diverter and  peristaltic pump. The diverter operates as an automatic switching valve to divert undesired portions of the eluate from the HPLC system to waste before the sample enters the MS.”

Line 123: Provide the g-force or rpm+rotor type for the centrifuge step, otherwise, it is not reproducible.

Thank you very much for this remark. All detailed information on the centrifugation conditions is provided below in 3.1.4. Optimization of soil extraction.

However, according to the reviewer suggestion we add information about rpm and rotor type:

“Soil extractions were carried out using an ultrasonic cleaner (Sonic 5, Polsonic, Poland), and then the samples were centrifuged using a Beckman Coulter Avanti JXN-26 centrifuge (20.000 rpm, JA-25.50 Fixed-Angle Aluminum Rotor type).”

Line 128: “acidified” with which acid?

Suprapur 65% Nitric acid was used for acidifying solutions.

I wrote with what acid acidified:

“Working standard solutions were obtained by appropriate dilution of the stock standard solutions using acidified (suprapur 65% nitric acid, Merck, Germany) Milli-Q-Gradient ultrapure deionized water (Millipore, Milli-Q-Gradient ZMQ5V001).”

Line 155: what is the difference between F3-A and F3-B? Were both methods used or only one of them?

In F3 fraction both methods (A and B) were used.

In  A) (10 ml H2O2 30% pH 2; 2 h water bath 85±2o C) - metal forms related to organic matter

  1. B) 25 ml 1 M NH4OAc pH 2; 16 h continuous mixing; (pH 2, supplied HNO3) - metal forms related to

The figure 1 and the corresponding results should be discussed in the results and discussion part. Also the title should be removed since the figure is explained in the caption. The y-axis should be named. The x-axis is not informative since the reader doesn’t know the meaning of the labels.

Figure 1 has been discussed in a new chapter in Results and discussion: 3.1.5 Sequential chemical extraction. Figure 1 has been improved.

The title has been removed.

Please check the numbering of tables and figures and their position in the text. It seems they were mixed up.

I checked the numbering of tables and figures and their position in the text.

Line 182: please clarify if you used the intensity values or the concentrations values for calculating the LOD. Usually, one uses three times the standard deviation of the intensity of a blank sample divided by the slope of the calibration curve. Furthermore, the determination of the LOD requires a high certainty on the slope, which is not provided with only three points. I recommend determining the LOD using low concentrations and more standards (e.g. 5 + a blank).

Thank you very much for this remark. Detailed information about this issue has been clarified in 3.1.6. Quality control of the speciation analysis

Limit of detection of tellurium species was determined based on the parameters of the calibration curve. For this purpose, a series of standard solutions Te(IV) and Te(VI) with a concentration of 1 µg, 10 µg / L, 25 µg / L were prepared. The limit of detection was determined using three times the standard deviation of the intensity of a test sample divided by the slope of the calibration curve according with the following formula:

LOD=(3,3×S)/b

where: LOD - limit of detection, S- standard deviation of the intensity of analyzed sample, b - the slope of the calibration curve. The standard deviation values were determined as a standard deviation for seven spiked with known amount of each tellurium species solutions.

The ICP-MS technique shows the linearity of the calibration function up to nine orders of magnitude of the signal, therefore a three-point calibration curve was chosen.

The description of the IC-method is missing (flow rate, eluent compositions and pH-values which were tested, injection volume, etc.). Also the data analysis (software, integration method, etc.) should be described.

Thank you very much for this remark. Information has been supplemented as recommended by the reviewer.

3.1.1 Elution optimization

During the research on separating tellurium ionic forms, the type of column (Hamil-ton PRP-X100 and Dionex IonPac AS7), concentration of eluents (6-10 mM Na2EDTA and 4-8 mM KHP), pH of eluents(in the range of 4.29 - 4.32), separation temperature (20-30 0C), method of preparation of the standards (water or 0.5 % HNO3), and complexing with complexing acids were optimized. Te(IV)/Te(VI) ions were separated with the Hamilton PRP-X100 column (150 mm x 4.6mm, 5µm). The anion exchange column Dionex IonPac AS7 (250 mm x 4 mm, 10 µm) was tested, but the obtained chromatogram presented wider peaks and stronger tailings of tellurium species (data not shown).

In general information about replication is missing.

Thank you very much for this remark. The text says several times that:

Each sample was measured three times using ICP-MS.

(2.5. Determination of the total tellurium and tellurium species content, Table 3).

Resuts and discussion:.

Lines 200-202: redundant with the method.

The sentence has been removed as information on this has been posted in methods.

Table 3 belongs to the method part.

Table 3 has been moved to methods.

For a better understanding of the separation process, it is important to discuss the electric charge of the Te-species at the different tested pH-values and how it influences these processes.

Thank you very much for this remark.

Figure 2: add a chromatogram of a soil sample for comparison. Why is the Te(IV) signal larger than the Te(VI) signal? The concentrations are supposed to be the same.

In reference to the reviewer's remark, we introduced a new Figure to the article showing the chromatogram of the real sample EDEL116 and the chromatogram of the sample of the 10 ppb standard. Te(VI) peak is wider and smaller in intensity, while peak Te(IV) is narrower and has more attention. The area of both peaks is comparable.

Figure 3: the data points are not visible on the version I received.

The chart has been posted again.

Line 253: Please clarify which extraction (BCR or for the IC-ICP-MS) is discussed.

 Prepared in accordance with the reviewer's comment.

3.1.4. Optimization of soil extraction for the IC-ICP-MS analysis.

Lines 253-261: this part belongs to the methods section. The conditions of the sonication and centrifugation steps should be explained in more details the methods section.

The conditions of the sonication and centrifugation steps in more details have been described the methods section.

Line 273: Was the total Te-content determined in the extract?

The total Te content was determined in soil extract and after total digestion. A single laboratory soil sample was divided and part of this sample was degsted (total Te content was quantified with ICP-MS) and extracted (total Te content in extract was quantified with ICP-MS). Figure 3 shows results of total tellurium content in soils extracts using various extractants.

Figure 3: Replace “extrahents” with “extractants”. Remove the title. Please replace the concentrations with the recovery values. This will be more informative for the reader. Add standard deviations on the graph.

Thank you very much for this remark. Te Figure has been improve.

The matrix effect should be tested using e.g. standard addition with more standards in order to compare the slopes in the matrix and in pure water. Or were the standards matrix matched? In

this case this should be mentioned in the method.

The matrix effect were tested using NaCl (effect of chlorides on separation) and sodium carbonate solution (effect of carbon on separation). No effect of chlorides and carbonates on the tellurium speciation analysis was observed. The relevant information was written in the text.

Lines 316-318: I disagree with this statement since (from the reader perspective) there are not enough information on the measured samples to discuss any effect of the plant on the soil pollution.

I understand the reviewer's remark and agree with him. The test results of a few samples are insufficient for such a definitive opinion. However, our preliminary research results indicate that in the area surrounding the electrowaste treatment and sorting facility, the concentration of tellurium was higher in the topsoil. In this publication, we focused mainly on presenting the proposed research methodology. Another paper is under preparation, in which we discuss in detail the research on almost 100 soil samples taken from soil cores (30 cm), taken at different distances from the emitter.

The sentence indicated by the reviewer was corrected.

“Our preliminary results indicate the influence of sorting and processing electrowaste plant on the increasing tellurium concentration in the topsoil of the surrounding areas.”

Table 4: Please discuss why the extraction efficiency is varying so strongly between samples (10-47%). Use µg/kg for better readability.

Soil samples were taken of various types. Some of them were forest soils, and some of them were typically urban anthropogenic soils. The extracts of these soils differed significantly, even organoleptically. Forest soil extracts were dark brown in color, and those from urban areas were straw-colored. e.g. soil samples with low extraction efficiency (105 EDEL,

107 EDEL, 110 EDEL, 112 EDEL) were typically forest soils, containing a lot of organic substances, and their extracts were dark brown in color. On the other hand, the sample extracts, e. g EDEL52, EDEL56, EDEL58, EDEL61 had a straw color, and the extraction efficiency of these soils was high.

This information was added to the discussion of the research results.

For the Te-speciation I recommend using an Eh-pH diagram for a clearer discussion. It has to be noted that the speciation was considered only from the thermodynamic point of view. Depending on the chemical form in which Te in entering the soil, metastable species could also be formed (e.g. colloids or as complex with SOM).

Thank you very much for this remark. The study of inorganic tellurium forms in soil extracts using hyphenated techniques, the IC-ICP-MS system, is an introduction to our analyzes. Currently, a study is being prepared on the content of selected TCEs (tellurium germanium, thallium) and their species in the soils surrounding the electrowaste processing plant, in which various geochemical, magnetometric, speciation, kinetic, thermodynamic and environmental aspects are analyzed. Therefore, in this work, I limited the scope of the discussed problems, limiting myself to the methodology. However, due to the reviewer's remark, an appropriate discussion has been included.

Lines 347-349: this contradicts the BCR-results where the concentration in the Fe/Mn-oxides phases was quite low.

Thank you very much for this remark.

In this excerpt, we present what other authors have reported in articles on tellurium fractionation in soils:

“In particular, a strong association has been observed between Fe3+ oxide minerals and tel-lurium. Te(VI) can be incorporated into the structures of Fe3+ oxides, whereas Te(IV) tends to be bound more weakly to Fe3+oxides by surface interactions only [42].”

Of course, an appropriate comment was added on this topic.

“However, as our research has shown (Figure 4), in the case of soil samples tested in the area surrounding the WEEE plant, tellurium was mostly associated with F3 (metal related to organic matter and sulphides) and R (residual) fraction. The share of tellurium bound to iron oxides in these soils did not exceed 10%.”

Round 2

Reviewer 1 Report

all corrections are made in accordance with the suggestions. I would recommend the paper to be accepted in the present form.

Author Response

Thank you very much for the review, valuable comments that helped to increase the value of this article.

Reviewer 2 Report

I thank the authors for their explanations and changes. I think that the manuscript is much clearer now. My last main concern is about the matrix effects, which should be checked more in details to my opinion, in particular, because the authors worked on samples which differ strongly from each others and some misinterpretations could occur if these effects are significant. See below for more detailed explanation. In addition, I have some minor remarks.

Introduction

Line 62: “mg/g” should be mg/kg or µg/g

Line 90: a verb is missing + I think this sentence would be better inserted in the next paragraph. For example: “Compared to hyphenated separation techniques, selective solid-phase extraction methods are interesting alternatives for samples with lower concentrations.” Indeed hyphenated techniques are generally less tedious due to reduced sample preparation steps.

Figure 1 and table 1: thank you for adding them, with the authors’ explanations, I now understand better the context of this research. However, the reader may be confused because there is no explanation on how the concentration map was produced. As I understood, more samples were sampled over the whole area and measured for Te but only the ones showed on the map were tested for speciation. I think this should be explained in the caption and mentioning the other study (even if not published yet) would be very useful for the reader.

Methods

Line 200: Mention the suppliers for the pump and diverter. In addition, provide the timing of the diverter. Which fraction was injected into the ICP-MS?

Line 202: “MS” should be “ICP-MS”.

BCI-Extraction: it is still unclear which method (F3A or B?) was used to make the figure 6 and discussed in the text. Please add the information in the caption or in the method description.

Line 265: Add the information on the chemicals in the “reagents” section and the used concentrations for the matrix effect.

2.51 and 3.1.6 are not consistent. Please update the method part.

Matrix effects: matrix effects should be tested using a spiked soil solution. Testing the effect of some ions separately does not take into account all possible sources of interference in soils such as SOM, for example. Therefore, I recommend to test matrix effect using the standard addition methods. The slope obtained in the sample should be close (can be tested with ANCOVA) to the slope of the calibration curve obtained with external standards, if matrix effects are negligible. It is particularly relevant for this study because, as the authors stated in their response, the tested samples were quite diverse in terms of soil composition. Therefore, it is important to prove that the soil composition is not affecting the method’s performances. Otherwise, standard addition has to be performed for each individual samples, which makes it much more tedious.

Figure 6: Correct for “percentage” or simply share in % and then remove the “%” signs in the y-axis.

Author Response

Response to editor’s and reviewers’ comments

Manuscript ID.: molecules-1170128

Title: Development of a Tellurium Speciation Study Using IC-ICP-MS on Soil Samples Taken from an Area Associated with the Storage, Processing, and Recovery of Electrowaste

Journal: Molecules

Article Type: Original research paper

# Reviewer comments in black

# Response in blue

Response to editor

Dear Ms. Jovana Curic, MSc, Assistant Editor, MDPI

We are submitting a revised manuscript (Molecules). We thank the editor and the reviewers for their suggestions. We are grateful for the helpful comments and advices from editors and reviewers. It is a valuable chance to improve our manuscript.

We have adjusted the manuscript and have provided a response for each of the comments and noted the changes made. We believe the new manuscript is more logic streamlined than the old one. For each change in the manuscript, the modified sentence(s) along with line numbers of changes in the revised version are provided, so that you could be easily identified. Minor revision included all Reviewer 2 comments. The changes are marked in manuscript (track).

We sincerely hope that our revised manuscript would meet the standards of publication.

Yours sincerely,

dr hab. Magdalena Jabłońska-Czapla, Prof. IPIS PAN

PhD

Institute of Environmental Engineering of Polish Academy of Science
Sklodowska-Curie 34 Street, 41-819 Zabrze, POLAND
tel. +48 32 2716481, fax. +48 32 2717470
e-mail: [email protected]

Reviewer #2:

I thank the authors for their explanations and changes. I think that the manuscript is much clearer now. My last main concern is about the matrix effects, which should be checked more in details to my opinion, in particular, because the authors worked on samples which differ strongly from each others and some misinterpretations could occur if these effects are significant. See below for more detailed explanation. In addition, I have some minor remarks.

Introduction

Line 62: “mg/g” should be mg/kg or µg/g

Thank you very much for this remark. Changes were made to the text in accordance with the reviewer's remark.

Line 90: a verb is missing + I think this sentence would be better inserted in the next paragraph. For example: “Compared to hyphenated separation techniques, selective solid-phase extraction methods are interesting alternatives for samples with lower concentrations.” Indeed hyphenated techniques are generally less tedious due to reduced sample preparation steps.

Thank you very much for this remark. Changes were made to the text in accordance with the reviewer's remark.

One sentence (without a verb) was replaced with: Compared to hyphenated separation techniques, selective solid-phase extraction methods are interesting alternatives for samples with lower concentrations.

Figure 1 and table 1: thank you for adding them, with the authors’ explanations, I now understand better the context of this research. However, the reader may be confused because there is no explanation on how the concentration map was produced. As I understood, more samples were sampled over the whole area and measured for Te but only the ones showed on the map were tested for speciation. I think this should be explained in the caption and mentioning the other study (even if not published yet) would be very useful for the reader.

Thank you very much for this remark. Figure 1 was prepared using Surfer 8 program on the basis of the samples chosen for Te speciation analysis and a total content as well (in the number of 14). On the map locations of analyzed soil samples were marked. In this area a total of 30 soil cores were collected, from which 66 soil samples were then subjected to geochemical analyzes (non-published data). These results will be subject of the next manuscript which will be prepared in the near future.

Figure 1 caption has been changed and relevant information has been added to the text.

Methods

Line 200: Mention the suppliers for the pump and diverter. In addition, provide the timing of the diverter. Which fraction was injected into the ICP-MS?

The diverter is used as a valve and switches automatically when the HPLC analysis starts and remains in this position until the entire sample series is completed.

Line 202: “MS” should be “ICP-MS”.

Thank you very much for this remark. I have made a correction.

BCI-Extraction: it is still unclear which method (F3A or B?) was used to make the figure 6 and discussed in the text. Please add the information in the caption or in the method description.

Thank you very much for this remark. The BCR soil extraction in F3 fraction F3A and F3B are combined. The entire F3 fraction is shown in the Figure 6.

I added the appropriate information in methods, in Figure 6's caption and in the results discussion.

Line 265: Add the information on the chemicals in the “reagents” section and the used concentrations for the matrix effect.

The information on the chemicals used for matrix effects cheked have been described in 2.3 Reagents section. Concentration of chemicals have been described in 3.1.6 section

2.5.1 and 3.1.6 are not consistent. Please update the method part.

Thank you very much for this remark. We are very sorry for this inconsistency. This resulted from the requests of Reviewer 1 to present quality control in this way.

These provisions have been unified.

The following changes have been made to the text:

2.5.1. Quality control of total tellurium concentration

Limits detection of the method of total analysis (LOD) was determined as three times the standard deviation value for the test sample witch were seven spiked samples (5-10 x noise) taken through all the sample preparation steps before and analyzed. The limits of quantification (LOQ) were expressed as three times the limit of detection value. The limit of detection and quantitation of tellurium was 0.006 mg/kg and 0.02 mg/kg, respectively. In order to validate the methodology of the total tellurium content, the certified reference material NCS DC 73324 (China National Analysis Center for Tron and Steel, Beijing, China) was used. The digested CRM sample was analyzed for tellurium content, yielding 105 % recovery of tellurium.

3.1.6. Quality control of the speciation analysis

Limit of detection of tellurium species was determined through measuring a series of standard solutions for two tellurium speciation forms. The linear model of the concentration dependence on total analyte counts was selected. Using the numerous determinations of the calibration curves, they also helped to calculate the limit of detection (LOD) for each form. The LOD calculation was based on the following dependence:

LOD=(3.3×S)/b

where: LOD - limit of detection, S- standard deviation value, b - the slope of the calibration curve.

The standard deviation values were determined as a standard deviation for seven sample solutions spiked with known amount of each tellurium species solutions. Limit of detection tellurium species Te(VI) and Te(VI) was 0.002 mg/kg and 0.004 mg/kg, respectively. The limits of quantification (LOQ) were expressed as three times the limit of detection value.

Due to the lack of certified reference materials containing both tellurium forms, the methodology for determining tellurium species was checked on the basis of certified reference material, which was extracted like the real soil samples. The obtained results (Table 4) showed that the CRM sample contain mainly a reduced tellurium form, and its recovery was 105 % (the concentration of tellurium from the certificate is 0.4 ± 0.1 mg/kg). Moreover, speciation analysis of the real soil extract (sample 105 EDEL) with the addition of 5 μg/L of the mixture of Te(VI) and Te(IV) standards was performed. The results showed tellurium content recovery of 111 % for Te(VI) and 80 % for Te(IV).

Matrix effects: matrix effects should be tested using a spiked soil solution. Testing the effect of some ions separately does not take into account all possible sources of interference in soils such as SOM, for example. Therefore, I recommend to test matrix effect using the standard addition methods. The slope obtained in the sample should be close (can be tested with ANCOVA) to the slope of the calibration curve obtained with external standards, if matrix effects are negligible. It is particularly relevant for this study because, as the authors stated in their response, the tested samples were quite diverse in terms of soil composition. Therefore, it is important to prove that the soil composition is not affecting the method’s performances. Otherwise, standard addition has to be performed for each individual samples, which makes it much more tedious.

Thank you very much for this remark it is very valuable to us. According to insights about matrix effects the article has been enriched with text about influence of the matrix and method of elimination used by us. The ANCOVA method offers many possibilities and information about samples properties and will certainly be used by us in research. We do not have enough data at the moment, but further research will be enriched about this information.

We have added a chapter on matrix effects in the text. In addition, four literature items were added.

The matrix interferences

Soil type can affect the performance of the method. Soil components, such as organic matter and minerals (clay, sand, silt) in varying proportions, affect the physical, mechanical, chemical and water properties of soils. Matrix effects mainly in recovery, and  increase or decrease the response during the analysis due to the presence of substances interfering with the detection of the target compounds. In general the effects of soil parameter on determination of tellurium species are poorly published. The research of Salvia et al. [1] based on the ANCOVA model shows that among other things organic carbon – being one of the main component of SOM (soil organic matter) had a significant impact on recovery. This is also evident in our research. Samples of higher organic matter content showed worse extraction efficiency. While it has been also reported that carbon- based compounds does not cause significant polyatomic interferences at masses 126Te and 128Te [2]. The research of Casiot et al. [3] on soils extract samples exposed to fulvic acid showed, that organic matter was not expected to interfere with the species of tellurium in the extract.

In our research due to minimalize spectral interferences coming from presence of Ba and Xe 125Te, 126Te, 128Te, 130Te isotopes were measured using correction equations (-0.003404*Xe129) for 126Te, (-0.072617*Xe129) for 128Te, and (-0.009437*Ba137-0.154312*Xe129) for 130Te. Toward compensate for matrix effects and drift in analysis of total concentration of elements, a technique with an internal standard was used. Standards, blanks and samples were measured using 103Rh as internal standard. Solution of 10 μg/L Rh was moved into all solutions and samples on line, by teeing in tubing on peristaltic pump.

The average concentration of Cl− and CO32­ in soil is 0.10 g kg-1 and about 0.5%, respectively [4]. Because of the large content of chloride and carbonate ions in soil as well as our earlier experience in speciation analysis study, it was decided to check the influence of Cl- and CO32-  ions.

The matrix effect were tested using NaCl solution (effect of chlorides on separation) and sodium carbonate solution (effect of carbon addition on separation).  The chloride interferences were checked by spiking real samples of soil extracts by 10 ug/L of sodium chloride solution. The same was done in the case of carbonate ions. No effect of chlorides and carbonates on the tellurium speciation analysis was observed. 

[1] Salvia, M.V.; Cren-Olivé, C.; Vulliet, E. Statistical evaluation of the influence of soil properties on recoveries and matrix effects during the analysis of pharmaceutical compounds and steroids by quick, easy, cheap, effective, rugged and safe extraction followed by liquid chromatography–tandem mass spectrometry. J. Chromatogr. A. 2013, 1315, 53-60. https://10.1016/j.chroma.2013.09.056

[2] Hu, Z.; Gao, S.; Günther, D.; Hu, S.; Liu, X.; YUAN, Y. Direct Determination of Tellurium in Geological Samples by Inductively Coupled Plasma Mass Spectrometry Using Ethanol as a Matrix Modifier. Appl. Spectrosc. 2006, 60, 781-5. https://doi.org/10.1366/000370206777887008

[3] Casiot, C.; Alonso, M.C.B.; Boisson, J.; Donard, O.F.X.; Potin-Gautier, M. Simultaneous speciation of arsenic, selenium, antimony and tellurium species in waters and soil extracts by capillary electrophoresis and UV detection. Analyst. 2008, 123, 2887-2893. https://doi.org/10.1039/A805954C

[4] Geilfus, C.M. Chloride in soil: From nutrient to soil pollutant. Environ. Exp. Bot. 2019, 157, 299-309. https://doi.org/10.1016/j.envexpbot.2018.10.035

Figure 6: Correct for “percentage” or simply share in % and then remove the “%” signs in the y-axis.

Thank you very much for this remark. The drawing has been corrected.
